# A major endogenous glycoside hydrolase mediating quercetin uptake in *Bombyx mori*

**Ryusei Waizumi** *, **Chikara Hirayama, Shuichiro Tomita, Tetsuya Iizuka, Seigo Kuwazaki, Akiya Jouraku, Takuya Tsubota, Kakeru Yokoi, Kimiko Yamamoto, Hideki Sezutsu**

Institute of Agrobiological Sciences, National Agriculture and Food Research Organization (NARO), Tsukuba, Ibaraki, Japan

* waizumir957@affrc.go.jp

**Data Availability Statement:** The raw genomic sequencing data of the Kosetsu strain are available in the Sequence Read Archive under the BioProject accession number PRJDB15366.

## Abstract

Quercetin is a common plant flavonoid which is involved in herbivore–plant interactions. Mulberry silkworms (domestic silkworm, *Bombyx mori*, and wild silkworm, *Bombyx mandarina*) take up quercetin from mulberry leaves and accumulate the metabolites in the cocoon, thereby improving its protective properties. Here we identified a glycoside hydrolase, named glycoside hydrolase family 1 group G 5 (GH1G5), which is expressed in the midgut and is involved in quercetin metabolism in the domestic silkworm. Our results suggest that this enzyme mediates quercetin uptake by deglycosylating the three primary quercetin glycosides present in mulberry leaf: rutin, quercetin-3-*O*-malonylglucoside, and quercetin-3-*O*-glucoside. Despite being located in an unstable genomic region that has undergone frequent structural changes in the evolution of Lepidoptera, GH1G5 has retained its hydrolytic activity, suggesting quercetin uptake has adaptive significance for mulberry silkworms. *GH1G5* is also important in breeding: defective mutations which result in discoloration of the cocoon and increased silk yield are homozygously conserved in 27 of the 32 Japanese white-cocoon domestic silkworm strains and 12 of the 30 Chinese ones we investigated.

## Author summary

Quercetin is one of the most abundant flavonoids present in plants. This flavonoid is involved in herbivorous insect–plant interactions. Insects utilize it for host–plant recognition, coloration, and protection from ultraviolet and oxidative stress. However, the molecular mechanism of quercetin metabolism in insects remains unclear. Mulberry silkworms (domestic silkworm, *Bombyx mori*, and wild silkworm, *Bombyx mandarina*) take up quercetin from mulberry leaves and sequester it into their cocoon to improve its protective properties. In this study, we identified an endogenous glycoside hydrolase in the domestic silkworm, named glycoside hydrolase family 1 group G 5 (GH1G5). This enzyme mediates quercetin uptake into the midgut cells by deglycosylating mulberry leaf-derived quercetin glycosides. This is the first discovery of a rutin glycoside hydrolase in an animal. Furthermore, we found that defective mutations of *GH1G5* have been broadly disseminated within the domestic silkworm population due to the improved cocoon color (i.e., discoloration to white) and increased silk yield. This study illuminates the unique

**Funding:** This work was supported by MAFF Commissioned project study on "Research project for sericultural bio-industry" Grant Number JP22680575 to R.W., T.I. and S.T., and JSPS KAKENHI Grant Numbers 21H03831 to K.Y. and H.S. The funders had no role in study design, data collection and analysis, decision to publish, or preparation of the manuscript.

**Competing interests:** The authors have declared that no competing interests exist.

mechanism of quercetin uptake in the domestic silkworm and uncovers an important event in the history of silkworm breeding.

## Introduction

Quercetin (3,3´,4´,5,7-pentahydroxyflavone) is a flavonoid abundantly found in a wide variety of plants [1]. Previous studies have found quercetin glycosides possess oviposition and feeding stimulant activity in lepidopteran, orthopteran and coleopteran insects [2,3,4,5,6]. These observations suggest that quercetin is widely ingested by insects. Quercetin ingestion by insects is likely not merely a consequence of the identification and feeding on host plants; it may have adaptive significance. In the common blue butterfly (*Polyommatus icarus*), which sequesters quercetin-3-*O*-galactoside in its wings, flavonoid content is higher in the female than in the male and positively correlated with female sex attraction [7,8,9]. The yellow pigment in the wings of a grasshopper (*Dissosteira carolina*), which may contribute to their camouflage in plants, results from sequestration of quercetin-3-*O*-glucoside (isoquercitrin, Q3G) [10]. Although these studies strongly emphasize the broad significance of quercetin in herbivore–plant interactions, the underlying molecular mechanisms of quercetin metabolism in insects are poorly understood.

Mulberry silkworms (domestic silkworm, *Bombyx mori*, and wild silkworm, *Bombyx mandarina*) accumulate various compounds derived from plants, including quercetin glucosides, kaempferol glucosides, and carotenoids, in their silk glands and colored cocoons (S1 Fig) [11,12,13]. Quercetin glucosides stored in the body and in the cocoon are reported to have antioxidant, ultraviolet-protective, and antibacterial properties [14,15,16]. Quercetin is found in the leaves of the mulberry tree (*Morus alba*), the sole food source of mulberry silkworms, as a series of glycosides formed by glycosylation at the 3-*O* position. The three most common quercetin glycosides in mulberry are quercetin-3-*O*-rutinoside (rutin), quercetin-3-*O*-malonylglucoside (Q3MG) and Q3G, which account for 71%–80% of the total flavonol content in the leaves [17].

The color of the cocoon of the domestic silkworm has been diversified through breeding, which indicates that the kinds and amounts of flavonoids that accumulate in the cocoons differ between strains [11,12]. Particularly, cocoons containing high flavonoid concentrations express a yellow-green color and are known as "green cocoons". Because accumulation in the cocoon is the end step of flavonoid metabolism, forward-genetic analysis focused on cocoon flavonoid content can reveal the genes involved in individual steps of flavonoid metabolism. Indeed, several loci associated with flavonoid metabolism have already been identified through this approach: the *Green b* locus, which encodes a uridine 5´-diphospho-glucosyltransferase (UGT) with a rare enzymatic activity glycosylating the 5-*O* position of quercetin [14], and the *New Green Cocoon* (*Gn*) locus, which encodes clustered glucose transporter (GLUT)-like sugar transporters presumed to import quercetin glucosides from the hemolymph to the silk gland [18]. However, these findings explain only a part of the process from quercetin uptake to the final accumulation of its metabolites in the cocoon. Elucidating the steps of flavonoid metabolism in the silkworm can provide valuable insights into the understanding of herbivore–plant interactions.

Here, we performed a quantitative trait locus (QTL) analysis focused on cocoon flavonoid content in the domestic silkworm and identified a novel locus, *Green d* (*Gd*), which is associated with this trait. Within the locus, we identified a glycoside hydrolase gene, glycoside hydrolase family 1 group G 5 (*GH1G5)*, which mediates quercetin uptake into the midgut cells by

deglycosylating mulberry leaf-derived quercetin glycosides. Genetic dissection of the novel gene revealed the contribution of the gene to improvement of the cocoon in a commercial context through breeding.

## Results

### QTL analysis identified a novel locus associated with flavonoid content in cocoons

To identify genes involved in quercetin metabolism by means of a forward-genetics approach, we prepared a green-cocoon strain (p50; alias: Daizo) and a white-cocoon strain (J01; alias: Nichi01). The two strains exhibited a distinct difference in cocoon color and flavonoid content (Fig 1A and 1B). A gradated range of cocoon colors in their $F_2$ intercross offspring implied that the genetic differences between the two strains associated with flavonoid content were composite (Fig 1C). Therefore, we conducted a QTL analysis, which allows for simultaneous identification of multiple genetic loci involved in a phenotype of interest. The flavonoid content of the cocoons of the $F_2$ population were scored by an absorbance-based method according to a previous report [19]. The QTL analysis was performed by using the phenotypic data of 102 individuals and 1038 genetic markers obtained from double-digest restriction-associated DNA sequencing data [20] (S1 Table). From the composite interval mapping, we identified three significant QTLs for cocoon flavonoid content on chromosomes 15, 20, and 27 (Fig 1C). Previous linkage studies suggested a locus, named *Green c* (*Gc*), associated with the yellow-green color of cocoons, is located at an unknown position on chromosome 15 [21,22]. The QTL on chromosome 27 was presumed to correspond to *Gn* [18]. Since no green cocoon-associated locus has yet been reported on chromosome 20, we named that locus *Green d* (*Gd*). The contribution of the *Gd* locus to cocoon flavonoid content was the second largest of the three QTLs, with a percentage of phenotypic variation explained by each QTL (PVE) value of 24.57% (Fig 1C and S2 Table). The 95% Bayes credible interval of the *Gd* locus was 7,980,189–10,504,065 bp. The nearest marker to *Gd* was located at 10,265,033 bp, with a logarithm of odds (LOD) score of 19.99. The PVE values of the QTLs on chromosomes 15 and 27 were 7.04% and 56.05%. Significant additive effects were detected between the QTLs on chromosomes 15 and 27 and between those on chromosomes 20 and 27, with corresponding PVEs of 1.93% and 5.47% (S2 Table).

### *Gd* locus contains a glycoside hydrolase family 1 gene cluster

Using the genome assembly and gene models of p50T [23], we found a cluster of nine genes encoding glycoside hydrolases (*KWMTBOMO12222–25*, *KWMTBOMO12227*, *KWMTBOMO12229*, *KWMTBOMO12230*, *KWMTBOMO12233*, and *KWMTBOMO12236*) on the *Gd* locus, which were annotated as glycoside hydrolase family 1 (GH1), according to the nomenclature of carbohydrate-active enzymes provided by CAZy [24] (Fig 1D). In mammals, a critical step in quercetin metabolism is deglycosylation of quercetin glucosides by lactase/phlorizin hydrolase (LPH), a member of GH1. LPH is expressed in intestinal epithelial cells and is anchored on the brush border membrane where it hydrolyzes flavonoid glycosides [25,26,27,28]. The resulting free quercetin aglycon is then passively absorbed into the intestinal cells due to its increased lipophilicity. We hypothesized that the *GH1* genes clustered within the *Gd* locus are involved in quercetin metabolism in the midgut lumen of the domestic silkworm, playing a role similar to that of LPH in mammals. In an inferred phylogenetic tree of all 21 GH1 proteins in *B. mori* and their homologous proteins in representative Holometabola insects (the fruit fly, *Drosophila melanogaster*; the honeybee, *Apis mellifera*; the red flour beetle, *Tribolium castaneum*), the glycoside hydrolases encoded in the *Gd* locus formed a distinct

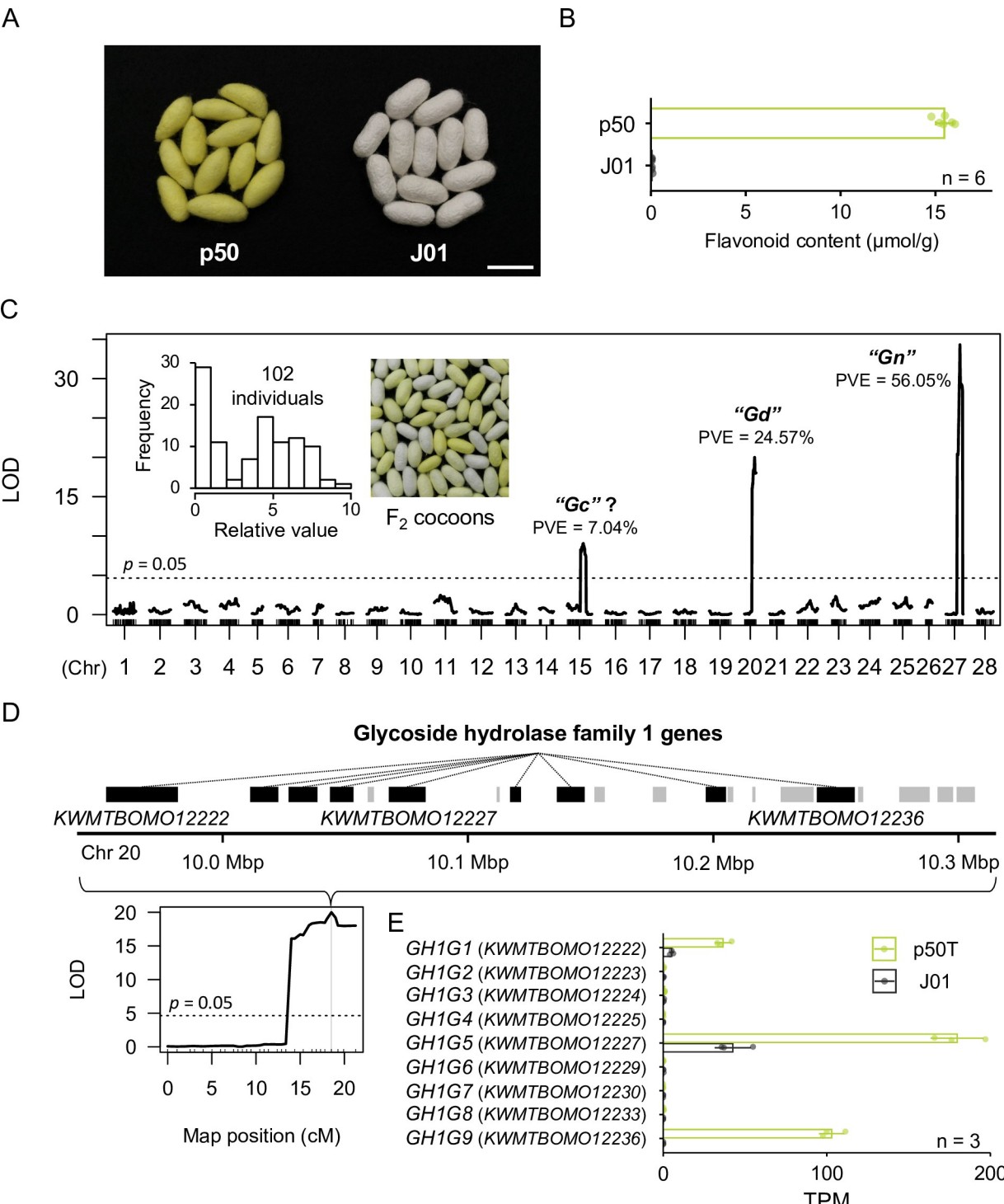

**Fig 1. Quantitative trait loci (QTLs) associated with cocoon flavonoid content.** (A) Photographs of representative cocoons of the p50 and J01 silkworm strains. Bar = 30 mm. (B) Flavonoid content of p50 and J01 cocoons. Data are means of n independent biological replicates ± SD. (C) QTL analysis for cocoon flavonoid content. The horizontal dotted line indicates the threshold of the permutation test (trials = 1000). Phenotype scores on the frequency distribution are relative to a maximum measurement of 10. LOD, logarithm of odds; PVE, percentage of phenotypic variation explained by each QTL. According to a previous study [21], the *Gc* locus is located on chromosome 15, but its detailed genetic or physical position is unknown. Therefore, it cannot be concluded that the QTL peak on chromosome 15 identified here corresponds to the *Gc* locus. (D) Gene models present within the *Gd* locus of the p50T genome assembly. The genes annotated as encoding glycoside hydrolase family 1 proteins are highlighted in black; their IDs are *KWMTBOMO12222, KWMTBOMO12223, KWMTBOMO12224, KWMTBOMO12225,*

*KWMTBOMO12227*, *KWMTBOMO12229*, *KWMTBOMO12230*, *KWMTBOMO12233*, and *KWMTBOMO12236* from the upstream side. (E) Expression of the candidate *Gd* genes in the midgut of third-day final instar male larvae. Data are means of n independent biological replicates ± SD. TPM, transcripts per million.

clade (group G), which was supported by a high bootstrap value (100%), suggesting that divergence of the group G-glycoside hydrolases had occurred after insect order divergence (S2 Fig). According to the phylogeny and their genomic positions, we named them as glycoside hydrolase family 1 group G 1–9 (GH1G1–9). The sequence identity among the group G glycoside hydrolase proteins was highest between GH1G2 and GH1G4 at 79.13%, and the sequence similarity was highest between GH1G1 and GH1G9 at 95.56% (S3A and S3B Fig). In addition, signal peptides were predicted in the N-terminal region of GH1G1, GH1G2, GH1G3, GH1G5, GH1G7 and GH1G9, suggesting they are secreted enzymes (S3A Fig). To identify which of them act in the midgut, we performed RNA-seq-based expression analysis. Three of the candidate genes, *GH1G1*, *GH1G5*, and *GH1G9*, were found to be strongly expressed in the midgut of p50T final instar larvae; the expression levels of these genes were significantly lower in the midgut of strain J01 (Fig 1E). Further examination of the expression profiles of the three genes with high expression in the midgut revealed that *GH1G5* and *GH1G9* were expressed specifically in the midgut, whereas *GH1G1* was expressed mostly in Malpighian tubules (S4 Fig). Taken together, we identified *GH1G1*, *GH1G5*, and *GH1G9* as candidate *Gd* genes which are involved in quercetin metabolism in the domestic silkworm.

## Functional analysis of the candidate *Gd* genes by CRISPR-Cas9

To investigate the involvement of these candidate genes in quercetin metabolism, we attempted to use a microinjection-mediated CRISPR-Cas9 system to establish p50T lineages in which they had been knocked out. Although we failed to establish knockout lineages for *GH1G1* and *GH1G9* due to the lethality of homozygous frame-shift mutations, we did manage to obtain two knockout lineages of *GH1G5* with different types of frameshift mutations in exon 5. We designated the one with a 5-bp deletion as Δ*Gd1* and the other with a 2-bp deletion as Δ*Gd2* (Fig 2A). The *GH1G5* mutations resulted in a premature stop codon at exon 5 along with shortened amino acid sequence lengths from 492 to 215 in Δ*Gd1*, and from 492 to 216 in Δ*Gd2*. The mutants produced discolored cocoons compared to p50T (Fig 2B). In addition, we observed a reduction of fluorescence under ultraviolet irradiation in the midgut, hemolymph, and silk glands of the mutants (Fig 2C–2E). Such fluorescence is characteristic of the accumulation of the two major quercetin metabolites in silkworm, quercetin-5-*O*-glucoside and quercetin-5,4´-di-*O*-glucoside [14,15,29]. Although knockout of *GH1G5* reduced the total flavonoid content in the cocoon to less than half that in the original p50T strain, it was still much larger than the effect predicted for the *Gd* locus in the QTL analysis (Fig 2F and S2 Table). We found similar reductions in the midgut, hemolymph, and middle and posterior silk gland. Because the flavonoid content in the cocoon differed largely between the insects reared with an artificial diet and those reared with fresh mulberry leaves (Figs 1B and 2F), we confirmed the flavonoid content reductions in the mutants in an experiment using fresh mulberry leaves (S5 Fig). Together, these results indicated that knockout of *GH1G5* resulted in malfunction of quercetin uptake into the midgut.

## Sequence and isoform determination for *GH1G5*

The predicted gene model for *GH1G5*, *KWMTBOMO12227*, consists of 11 exons, encoding a total of 492 amino acids. The accuracy of the predicted sequence and its dominance among isoforms were confirmed using Sanger sequencing and the RNA-seq data obtained from final

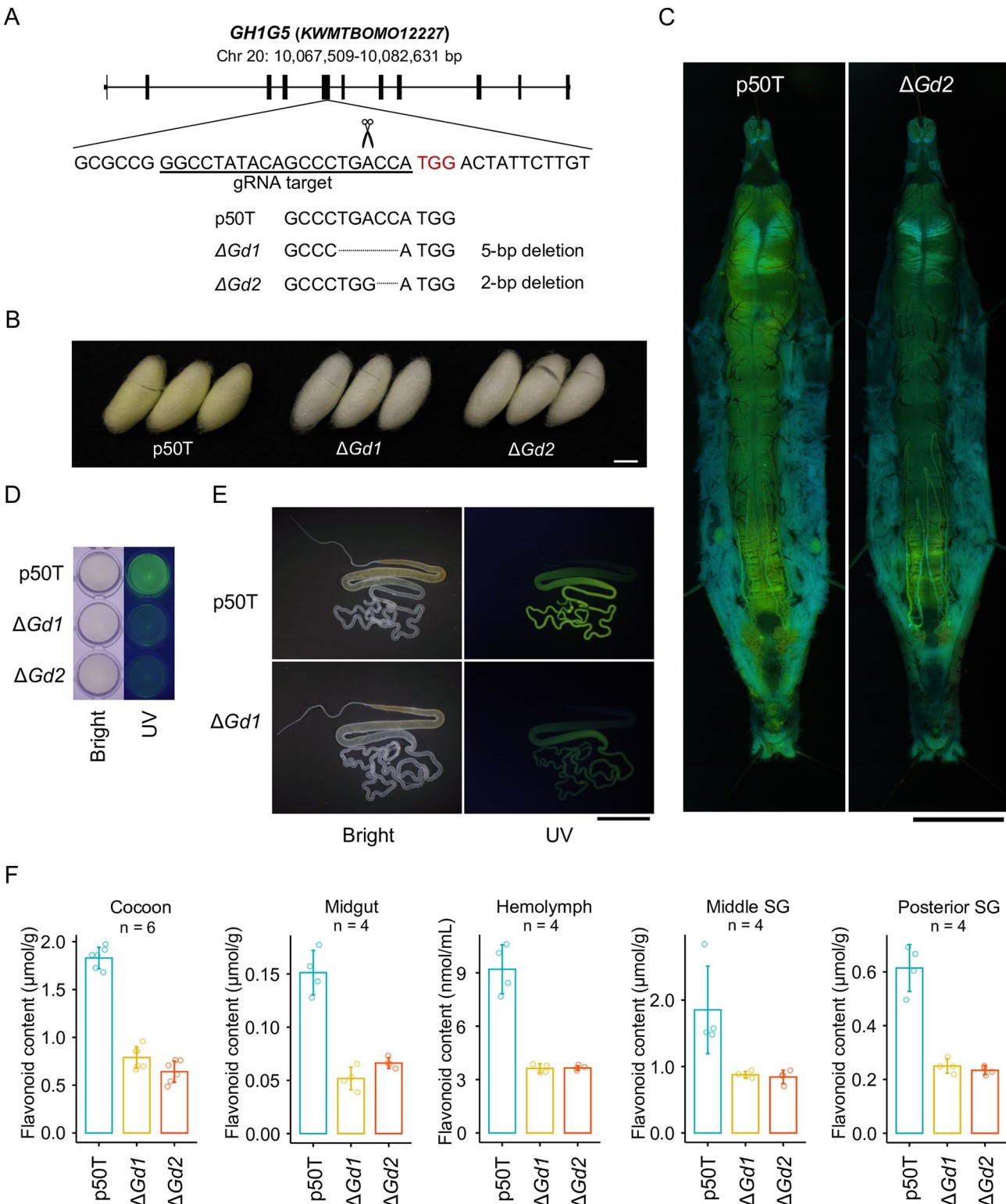

**Fig 2. Reduction of flavonoid content in *GH1G5*-knockout mutants.** (A) Disrupted target sequences in *GH1G5*. (B)–(E) Cocoons (B), midguts (C), hemolymph (D), and silk glands (E) of the p50T strain and the *GH1G5* mutants. Organs and tissues were taken from sixth-day final instar female larvae. Bars = 10 mm. (F) Total flavonoid content of cocoons, organs, and tissues of the p50T strain and the *KWMTBOMO12227* mutant lineages. Data are means of n independent biological replicates ± SD. All insects were reared on a commercial artificial diet. SG, silk gland.

instar larvae of strain p50T. We cloned the putative longest open reading frame in the transcripts of *GH1G5*, and sequenced it by Sanger sequencing (S6A Fig). The determined sequence was completely identical to the predicted sequence of *GH1G5*. Previously, our research group reported RNA-seq data for final instar larva tissues of the p50T strain [30]. By mapping the reads of the data derived from the midgut to the genomic sequence of the p50T strain reported by Kawamoto et al. [23], we determined the transcript isoforms of *GH1G5*. The gene model accurately represented the dominant isoform; another isoform extending 15-bp downstream of the 10th exon was also detected (S6B Fig). The termination codon in the extended region shortened the predicted protein sequence length of this isoform to 456 amino acids. Since the mean coverage of each base of the 15-bp extended region was only about 15% of that of the original 10th exon of *GH1G5* (S6C Fig), we concluded this to be a minor isoform.

## GH1G5 mediates quercetin uptake by hydrolysis of quercetin glycosides

Together, our knockout analysis and sequence characterization suggested that GH1G5 is secreted into the midgut lumen and mediates the uptake of mulberry-derived quercetin by deglycosylation of quercetin glycosides. However, the knockout mutants still accumulated some flavonoids in the midgut cells (Fig 2F). This might be due to the diversity of quercetin glycosides in mulberry leaves and the substrate specificity of GH1G5. To confirm whether GH1G5 is involved in deglycosylation of quercetin glycosides, we investigated the hydrolytic activity of midgut tissue of the knockout mutants on the three major quercetin glycosides in mulberry leaf: rutin, Q3MG, and Q3G (Fig 3A).

The homogenate of the p50T midgut exhibited hydrolytic activity against all three types of quercetin glycoside (Fig 3B). However, the activities were decreased in the knockout mutants, indicating that GH1G5 is involved in the hydrolysis of all three quercetin glycosides. Notably, *GH1G5* knockout completely abolished the hydrolytic activity against rutin. Furthermore, partial reductions were observed in the hydrolytic activity of the other two glycosides, Q3MG and Q3G, suggesting that the silkworm has other glycoside hydrolases with hydrolytic activities against those molecules (Fig 3B). Knocking out *GH1G5* reduced the hydrolytic activity of midgut homogenates against Q3MG and Q3G by 0.6 to 0.8-fold and 0.7 to 0.8-fold, respectively. These results suggested that GH1G5 is an important protein for the hydrolysis of quercetin glycosides in the silkworm lumen.

In 1972, Fujimoto and Hayashiya reported that the domestic silkworm accumulates flavonoids in its cocoon when reared on artificial diets containing isolated quercetin or rutin [31]. To investigate whether the uptake of rutin-derived quercetin is dependent on deglycosylation by GH1G5, we reared insects on semi-synthetic diets supplemented with rutin or quercetin but without mulberry leaf powder and measured the flavonoid content in the cocoons. The p50T strain accumulated flavonoids in its cocoon irrespective of diet (Fig 3C and 3D and S3 Table). Although the *GH1G5*-knockout mutants accumulated the same amount of flavonoids as did p50T when reared on the quercetin diet, they accumulated only 6% of that accumulated by p50T when reared on the rutin diet (Fig 3C and 3D). These results were consistent with the report by Fujimoto and Hayashiya, and suggested that the uptake of quercetin from rutin is strongly dependent on deglycosylation by *GH1G5*.

## Evolution of GH1G5

The phylogeny of the Holometabola GH1 proteins suggested that the divergence of the group G glycoside hydrolases had occurred after insect order divergence (S2 Fig). To estimate when *GH1G5* arose during the evolution of Lepidoptera, we constructed a maximum likelihood-inferred phylogenetic tree of lepidopteran-wide orthologous proteins of the group G glycoside

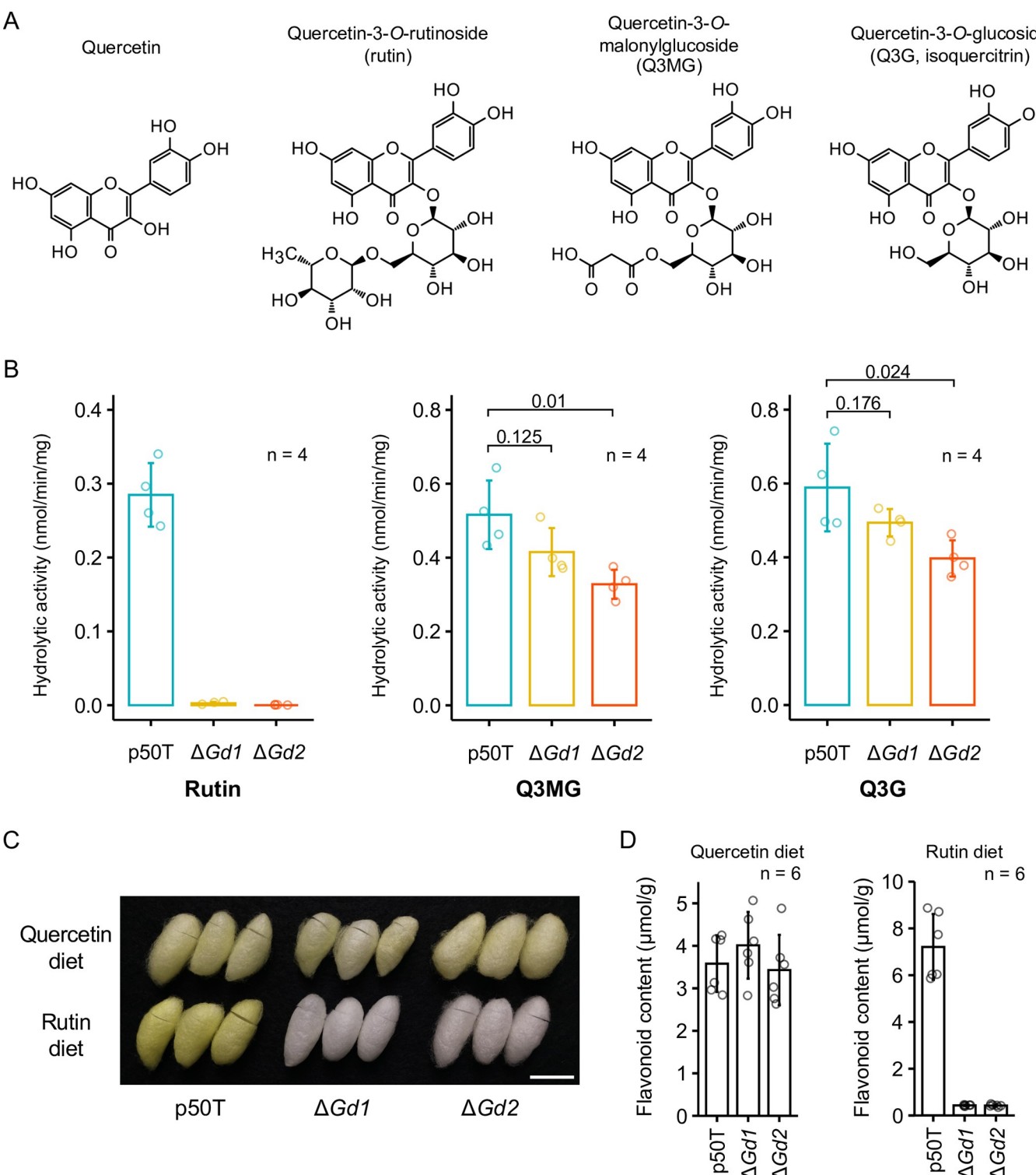

**Fig 3. Hydrolytic activity of GH1G5 on the three major quercetin glycosides in mulberry leaf.** (A) Structural formulae of quercetin and the three major quercetin glycosides present in mulberry leaf. (B) Hydrolytic activity of the midgut on the quercetin glycosides. Data are means of n independent biological replicates ± SD. The values above the graph are *p*-values calculated by a two-tailed Student's *t*-test. All samples were from fifth-day final instar female larvae reared on a commercial artificial diet. (C) Photograph of representative cocoons of the p50T strain and the mutants reared on semi-synthetic diets containing quercetin or rutin. Bar = 20 mm. (D) Total flavonoid content in cocoons of the p50T strain and the mutants reared on semi-synthetic diets containing quercetin or rutin. Data are means of n independent biological replicates ± SD.

hydrolases in *B. mori* (BmorGH1G1–9) (Fig 4A). The tree classified BmorGH1G1–9 into five clades: one clade including BmorGH1G5, one clade including BmorGH1G1 and BmorGH1G9, one clade including BmorGH1G2, BmorGH1G3 and BmorGH1G4, one clade including BmorGH1G6 and BmorGH1G7, and one clade including BmorGH1G8. The clade including BmorGH1G5 consisted of only proteins from species belonging to Macroheterocera (including Noctuoidea, Lasiocampoidea, and Bombycoidea), suggesting that *GH1G5* had evolved through gene duplication after the divergence of Pyraloidea and Macroheterocera (Fig 4A and 4B). Sequence identities and similarities of the GH1G5-class proteins, excluding the one from *B. mandarina*, to BmorGH1G5 were at a maximum of 57.92% and 89.98% (Fig 4C). OrthoFinder [32], the tool we used for the collection of orthologous proteins of BmorGH1G1–9, detected gene duplication events that had occurred within each orthologous group while simultaneously estimating orthologous relationships. Interestingly, it suggested that the group G glycoside hydrolases had undergone notably frequent duplication events; the estimated number of duplications which occurred within the ortholog group including GH1G5 (also including GH1G2, GH1G4, and GH1G6–9) was 46, which ranked 197th out of 19436 total groups and 1st out of 54 groups including *B. mori* proteins annotated as glycoside hydrolases (InterPro entry: IPR001360 or GO annotation: 0016798) (S4 and S5 Tables).

## Defective structural mutations of *GH1G5* were broadly disseminated in white-cocoon domestic silkworm strains

Although accumulation of carotenoids or flavonoids in the cocoon produces a variety of cocoon colors, it is white-cocoon strains that are the most popular in commercial use. Earlier in the present study, we found that defective mutations of *GH1G5* resulted in impaired quercetin absorption, which made cocoon colors closer to white (Fig 2B). Interestingly, the defective mutation of *GH1G5* did not impair the growth of the silkworm, but rather improved silk yield (S7 Fig). The implication here was that this beneficial mutation was selected for the establishment of white-cocoon domestic silkworm strains.

To examine our hypothesis, we determined conserved *GH1G5* genotypes in a collection of *B. mori* strains, and investigated their frequency and distribution. We first determined the details of the mutation introduced in J01, and then compared them to p50T. Previously, we reported a genome assembly and gene model for J01 [34]. A BLASTp search of our J01 gene model identified predicted gene *BMN13127* as corresponding to *GH1G5* in this strain. Comparing the genomic regions of *GH1G5* in p50T and J01, an insertion of 3997 bp was found in exon 5 of J01 *GH1G5* (Fig 5A). The inserted sequence was identified as the reported non-autonomous transposable element in *B. mori*, *Neet* [35]. The *Neet* sequence and the intronic region between exon 5 and 6 in J01 included the inverted sequence of the intronic region between exon 5 and 6 of *GH1G5* of p50T (S8A Fig), suggesting that it had experienced additional structural changes. A product amplified from the genomic DNA region of exon 5 of J01 including the insertion was approximately 4000-bp longer than that from the p50T genomic DNA (Fig 5B). Probably due to the insertion, the predicted splice site of the exon 5 end shifted 123 bp forward in J01 *GH1G5*, resulting in a deletion of 41 amino acids in the predicted protein sequence (Figs 5A and S8B). By cloning and sequencing the J01 transcript using the primer sets that were used to determine the open reading frame sequence of p50T *GH1G5*, the accuracy of the predicted sequence was confirmed (S8C Fig). The deletion included two of nine residues with >99% conservation across 101 homologous proteins from five Macroheterocera species, and all of seven residues which harbored strongly similar physicochemical properties across all of these proteins (S9 Fig). These observations strongly suggested that the deletion was mainly responsible for the observed difference in flavonoid content in the cocoon.

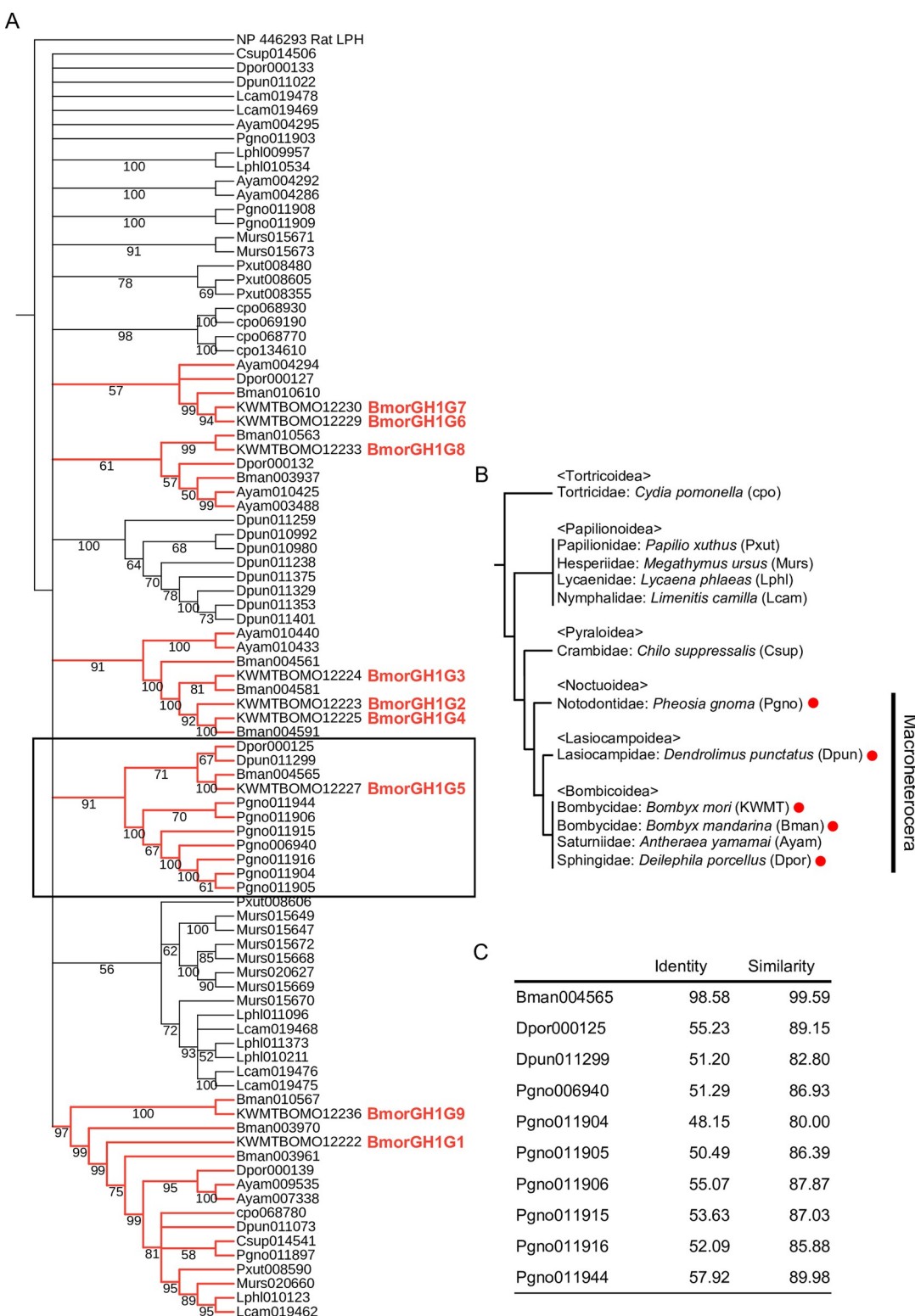

**Fig 4. Phylogenetic tree of lepidopteran-wide orthologous proteins of the group G glycoside hydrolases in *B. mori*.** (A) We constructed the phylogenetic tree by the maximum likelihood method. Values on the nodes represent the bootstrap scores (trials = 100). Unreliable nodes with bootstrap values under 50 are shown as multi-branching nodes. The tree was rooted using Rat LPH as an outgroup. Red branches represent clades including the group G glycoside hydrolases in *B. mori*. The clade containing BmorGH1G5 is highlighted with a box. (B) Species tree of Lepidoptera drawn/constructed with reference to the

report by Kawahara et al. [33]. Red circles indicate species harboring a GH1G5 ortholog. (C) Sequence identities and similarities of the GH1G5-class proteins to BmorGH1G5. Similarity of amino acid residue property is determined according to the groups of strongly similar properties described at Clustal Omega FAQ (https://www.ebi.ac.uk/seqdb/confluence/display/THD/Help+-+Clustal+Omega+FAQ).

Next, we examined the presence of the J01-type 4-kbp insertion in 67 Japanese and Chinese local strains by PCR using tested primers (S10 Fig and S6 Table). All five green-cocoon strains lacked the insertion. Of the 32 Japanese local white-cocoon strains, the insertion was absent in five and present in seven, whereas the remaining 20 lacked any amplification product. Of the 30 Chinese local white-cocoon strains, the insertion was absent in 15, present in four (p50T-type products were also observed in three of these strains), and 11 again lacked an amplification product. The lack of an amplification product suggested these strains contained another dysfunctional mutation. To determine what it might be, we obtained Illumina whole-genome sequencing data of Kosetsu, one of the strains lacking an observed amplification product, and assembled it. By comparison with the genomic sequence of p50T, a 143.7-kbp deletion including *GH1G5* was found in the Kosetsu *Gd* locus (Fig 5C). When the region including the deletion was amplified by PCR using p50T, J01, and Kosetsu genomic DNA, an observable amplification product was synthesized only from the Kosetsu sample (Fig 5D). In addition, no observable amplification product was synthesized from the Kosetsu sample by using reverse-transcription PCR to amplify the full-length open reading frame of *GH1G5* using cDNA from the midgut of sixth-day final instar larvae of the three strains (Fig 5E). Moreover, the J01-derived product was shorter in size and the band appeared fainter than that from p50T. These results were consistent with the genetic differences of *Gd* among the three strains predicted from the assembled genomic sequences. Using the primer sets for detecting the Kosetsu-type large deletion, we examined the presence of the mutation in all of the prepared 67 silkworm strains. It revealed that 46 of the 67 strains possessed the Kosetsu-type deletion. Genotypes of 26 of the 20 Japanese and 11 Chinese local white-cocoon strains which we failed to genotype were determined as the Kosetsu-type, whereas the remaining five still lacked any amplification product (S10 Fig). The five strains which we failed to genotype using any of the tested primer sets might have undergone further genomic structural changes after acquiring the Kosetsu-type large deletion.

To summarize the genomic PCR analysis, we found three haplotypes of *GH1G5* in the population: p50T-type (P, functional), J01-type (J, harboring a 4-kbp insertion in exon 5), and Kosetsu-type (K, harboring a large deletion eliminating *GH1G5*). The proportions of strains homozygous for the functional haplotype (P), heterozygous for the functional and a dysfunctional (J or K) haplotype, and homozygous for a dysfunctional haplotype were 3%, 13%, and 84% in the Japanese local white-cocoon strains, and 23%, 37%, and 40% in the Chinese local white-cocoon strains (Fig 5F).

## Discussion

### GH1G5 mediates quercetin uptake in *Bombyx mori*

Here, we identified GH1G5 as a glycoside hydrolase that initiates quercetin metabolism in the domestic silkworm. GH1G5, a putative secreted enzyme belonging to GH1, is expressed specifically in the midgut (S4 Fig). Knockout of *GH1G5* reduced total flavonoid content in the midgut, hemolymph, and silk glands, indicating that the gene is involved in quercetin absorption in the midgut, the initial place of entry into the insect (Fig 2C–2F). The hydrolytic activity of the midgut tissue on rutin, Q3MG, and Q3G was reduced in *GH1G5*-knockout mutants (Fig 3B), suggesting that GH1G5 hydrolyses all three major quercetin glycosides present in

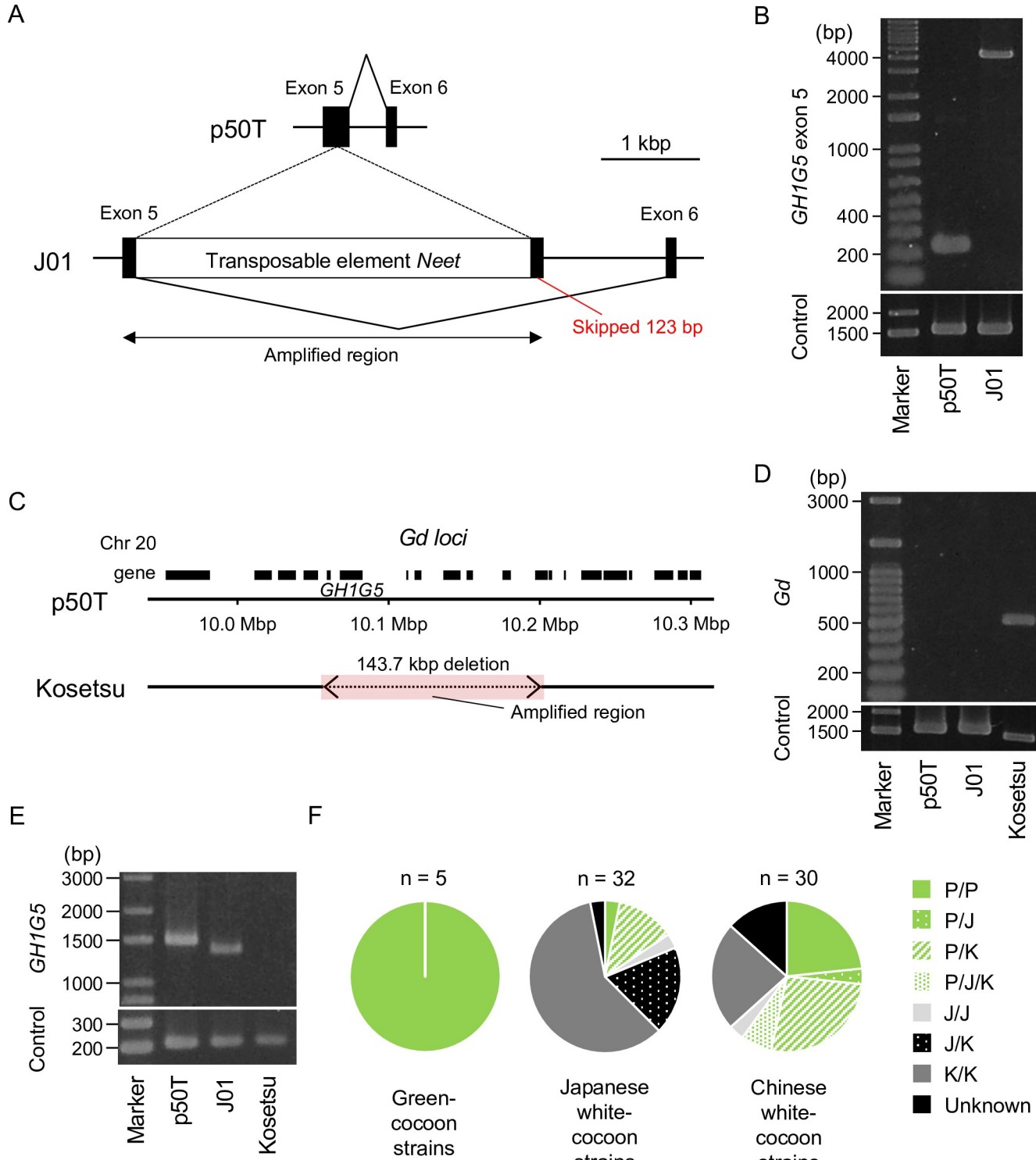

**Fig 5. Conserved structural mutations of *GH1G5* in the domestic silkworm population.** (A) Schematic illustration comparing exons 5, 6 and the intronic region between them (intron 5) of *GH1G5* from the p50T and J01 strains. "Amplified region" indicates the region amplified in (B). (B) PCR identifying the J01-type insertion. The primer set amplifying the genomic region from exons 1 and 2 of *KWMTBOMO14639* (*rp49*) was used as the control (predicted fragment length = 1548 bp). (C) Schematic illustration comparing the *Gd* loci of the p50T and Kosetsu strains. Black boxes indicate the predicted genes. The colored region indicates the region amplified in (D). (D) PCR identifying the Kosetsu-type large deletion. (E) RT-PCR of the *GH1G5* full-length open reading frame using cDNA libraries constructed from the midguts of sixth-day final instar larvae. The primer set targeting exons 1 and 2 of *KWMTBOMO14639* (*rp49*) was used as the control (predicted fragment length = 213 bp). (F) Distribution of the three haplotypes of *GH1G5*: p50T-type (P,

functional), J01-type (J, harboring a 4-kbp insertion in exon 5), and Kosetsu-type (K, harboring a large deletion eliminating *GH1G5*). P/P, only the p50T genotype was detected; P/J/K, p50T, J01, and Kosetsu genotypes were detected; J/J, only the J01 genotype was detected; J/K, both J01 and Kosetsu genotypes were detected; K/K, only the Kosetsu genotype was detected; U, unknown, nothing detected. Grayscale colors indicate pairs of loss-of-function haplotypes.

mulberry leaf. The amount of flavonoids contained in the cocoon of insects reared with a diet supplemented with rutin was greatly reduced in the absence of functional GH1G5 (Fig 3C and 3D), suggesting that deglycosylation of quercetin glycosides by GH1G5 is a critical step in quercetin uptake in the silkworm. The absorbed quercetin may be retained in midgut cells by decreasing its lipophilicity through 5-*O*-position glucosylation by UGT [14]. The present analyses were limited to studying functional changes in deletion mutants produced by genome editing and spontaneous mutations found by screening diverse domestic silkworm strains. Further biochemical and/or immunological studies will be needed to clarify detailed enzymatic characteristics and biological roles of GH1G5.

While revealing a major role for GH1G5 in quercetin uptake in the silkworm, this study also suggests the presence of GH1G5-independent quercetin uptake pathways. The accumulation of flavonoids in the cocoon and tissues was not completely lost in the *GH1G5*-knockout mutants (Fig 2F). In addition, the residual hydrolytic activity of the midgut homogenate on Q3MG and Q3G implies the presence of other glycoside hydrolases mediating quercetin uptake (Fig 3B). Although we were unsuccessful in obtaining knockout lineages for *GH1G1* or *GH1G9*, the proteins they encode may correspond to these other glycoside hydrolases. Interestingly, Q3G is observed in the midgut cells [12,14,15]. Given that some Gn_Strs are expressed in the midgut [18], the unknown glycoside hydrolases possibly hydrolysis imported Q3G in the cytoplasm.

In Fig 6, we present a model of quercetin metabolism in the silkworm, inferred from this study and relevant previous ones. This model illustrates that the studies identified only a part of the genes involved in the metabolic process. Although Gn_Strs certainly play a crucial role in transporting quercetin glucosides in the silkworm, it remains unclear whether they import quercetin glucosides into the cells, or export them out of the cells, and where these occur [18]. In mammals, GLUTs, the homologous transporter proteins to Gn_Strs, primarily facilitate sugar uptake from the extracellular environment [36], suggesting that Gn_Strs import quercetin glucosides into the cells of silk glands or midgut. However, in the small intestine, GLUT2 either transports glucose from the lumen into the cells, or from the cells to the circulation, depending on the sugar environment [37]. Considering this, it is possible that some Gn_Strs also export quercetin glucosides from the cells. Additionally, UGT-mediated glucosylation at the 5-*O* position in the midgut cells is not the sole modification that quercetin undergoes after absorption. A major form of quercetin glucosides in the midgut cells is quercetin-5-*O*-glucoside, while in the hemolymph, silk glands and cocoon, it is quercetin-5,4′-di-*O*-glucoside [14,15]. In some silkworm strains, the variety of quercetin glucosides expands within the silk gland cells, synthesizing quercetin aglycone and its glucosides which are glucosylated at the 3, 3′, 4′, or 7-*O* positions [12,14,15]. These observations suggest the presence of glycoside hydrolases and glucosyltransferases mediating the syntheses. Future research is anticipated to elucidate the overall molecular mechanisms of quercetin metabolism in the silkworm, providing a reference for flavonoid metabolism pathways in herbivorous insects.

## Adaptive significance of quercetin uptake in mulberry silkworms is supported by the fact that they have retained *GH1G5*

Our results indicate that deglycosylation-dependent flavonoid uptake, which has been well-studied in mammals, is also found in insects. One feature distinguishing the mechanism in the

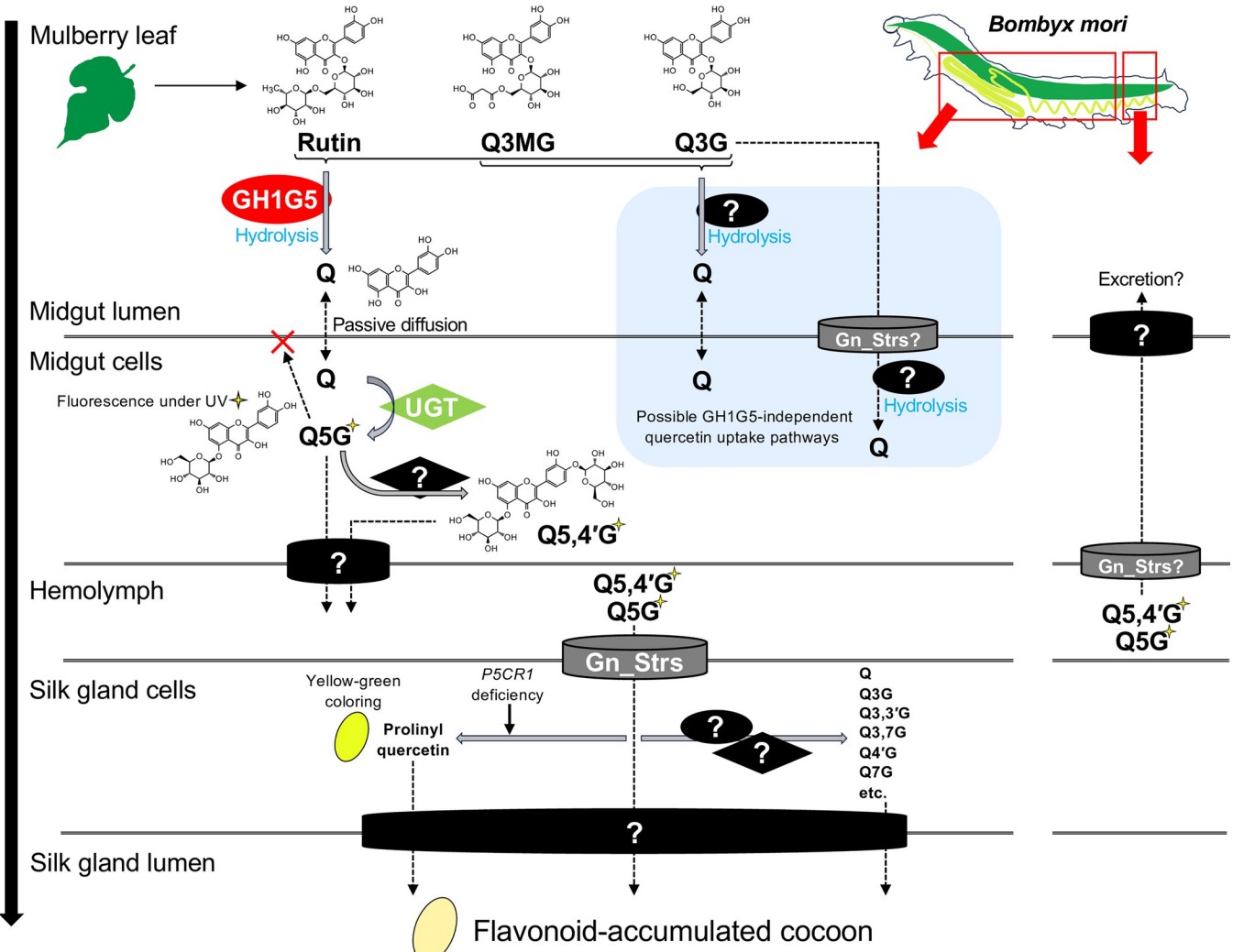

**Fig 6. Proposed model of quercetin metabolism in the domestic silkworm.** Q3MG, quercetin-3-*O*-malonylglucoside; Q3G, quercetin-3-*O*-glucoside; Q, quercetin; Q5G, quercetin-5-*O*-glucoside; Q5,4′G, quercetin-5,4′-di-*O*-glucoside; Q3,3′G, quercetin-3,3′-di-*O*-glucoside; Q3,7G, quercetin-3,7-di-*O*-glucoside; Q4′G, quercetin-4′-*O*-glucoside; Q7G, quercetin-7-*O*-glucoside. Glycoside hydrolases are represented as ovals, glucosyltransferases as diamonds and transporters as cylinders. Some domestic silkworm strains that accumulate flavonoids in their cocoons, including p50 and p50T, produce yellow-green cocoons known as "green cocoons". The color is derived from prolinylflavonols, the synthesis of which is promoted by deficiency of *pyrroline-5-carboxylate reductase 1* (*P5CR1*) [38]. Although the glucosylation of quercetin at the 5 or 4′-*O* positions of quercetin is illustrated here as occurring in the midgut, the glucosylation activity is also observed in the fat body, hemolymph and silk glands [14]. Similarly, glucosylation activities at other positions of quercetin are not exclusively observed in the silk glands [14].

domestic silkworm from that in other animals is the suggested rutin hydrolytic activity of GH1G5 (Fig 3B–3D). Rutin hydrolysis in mammals is dependent on the intestinal microbiota [39,40]. Rutin glycoside hydrolases have previously been identified in bacteria, fungi, and plants [41,42,43], but to our knowledge, not in animals. Rutin hydrolysis in the honey bee (*Apis mellifera*) also depends on the gut microbiota [44], indicating that endogenous rutin glycoside hydrolases are not common in insects. Rutin is a common plant flavonoid, accounting for 14%–46% of flavonol glycosides in mulberry leaf [17]. The accumulation of flavonoids was severely restricted in the cocoons of *GH1G5*-knockout mutants reared on a rutin-containing diet, suggesting the absence of rutin digestion pathways other than deglycosylation by GH1G5 (Fig 3C and 3D). Therefore, the acquisition of rutin hydrolytic activity should result in a

considerable change in quercetin intake. In addition, the genomic instability of the *Gd* locus emphasizes the significance of quercetin bioavailability brought about by the enzyme. According to the inference of gene duplication events by OrthoFinder [32], the group G glycoside hydrolases are the group that has undergone the most frequent gene duplications among the 54 *B. mori* glycoside hydrolase ortholog groups (S4 and S5 Tables). The fact that the large deletion was introduced into the *Gd* locus within a short period of the breeding history of the domestic silkworm also suggests the genomic instability of the region. While the genomic instability of the *Gd* locus had possibly provided genetic material to evolve *GH1G5* by gene duplication events, it can be inferred to have repeatedly exposed *GH1G5* to genomic structural changes which introduce defective mutations into the gene. However, mulberry silkworms have retained *GH1G5* and its rutin hydrolytic activity until defective mutations were selected in breeding for cocoon improvement. These results support that the suggested positive properties of quercetin glycosides, notably, their antioxidant [15], ultraviolet-protective [14], and antibacterial functions [16], improve the fitness of the wild silkworm in the natural environment. Although, it should be noted that excessive quercetin uptake is still toxic even for the domestic silkworm [45], meaning that mulberry silkworms should have a mechanism to maintain appropriate levels of quercetin. The fluorescence in the posterior region of the midgut observed in the present study implies such mechanisms may involve the accumulation of quercetin glucosides with glycosylation at the 5-*O* position and reloading of excessive quercetin glucosides from the hemolymph for excretion into the midgut lumen (Figs 2C and 6).

The phylogeny for the lepidopteran orthologous proteins of the group G glycoside hydrolases suggested that *GH1G5* evolved after the divergence of Pyraloidea and Macroheterocera (Fig 4A and 4B). This indicates that quercetin uptake mediation by GH1G5 is not common in lepidopteran insects. Given the low sequence identities (<58%) between BmorGH1G5 and the GH1G5-class proteins in other Macroheteroceran moths excluding *B. mandarina* (Fig 4C), the rutin hydrolytic activity might be conserved only in mulberry silkworms. Although we did not investigate rutin hydrolytic activities of the GH1G5-class proteins other than BmorGH1G5, such an investigation could highlight the distinctiveness of the enzymatic property of BmorGH1G5. If so, it would allow for a discussion on the association between the evolution of the rutin hydrolytic activity and the monophagy of the silkworms on mulberry leaves, which contain high levels of rutin [17]. Quercetin uptake is observed in other herbivorous insects [9,10]; their GH1 proteins may have evolved to adapt to flavonoid compositions in their respective host plants. A species-wide characterization of GH1 proteins in flavonoid glycoside hydrolytic activities would potentially provide insights into dietary habits of herbivorous insects.

### *GH1G5* loss contributed to cocoon improvement

Through breeding, many strains of the domestic silkworm have been established which exhibit a wide variety of cocoon colors depending on their flavonoid and carotenoid content. Nevertheless, white cocoons tend to be preferred commercially in regions such as Japan and China, especially since the 20th century. In a previous study that conducted a worldwide genetic analysis of silkworm strains, 91 of the 121 strains examined were white-cocoon strains [46]. The defective mutation in *GH1G5*, which manifests as reduced cocoon color (Fig 2B), may have contributed to the establishment of these white-cocoon silkworm strains. Of the white-cocoon strains examined in the present study, only 63% carried homozygous defective haplotypes of *GH1G5* (haplotype J or K; Figs 5F and S10 and S6 Table). In 24% of the strains, a mixture of defective and functional haplotypes (haplotype P) was present. These are not surprising results, considering that not all white-cocoon strains harbor multiple loss-of-function mutations of

the sugar transporters encoded in the *Gn* locus, which is more effective than *Gd* with respect to cocoon discoloration [18]. Even if visually determined as white, the actual color tone and therefore the flavonoid content of the cocoon varies from strain to strain [47]. In addition to *Gd*, *Gb*, and *Gn*, within each of which a gene involved in the quercetin metabolism has been identified, several other loci associated with cocoon greenness have been reported, including *Gc* [21,22], *Green cocoon* (*Grc*), *Green egg shell* (*Gre*), and *Yellow fluorescent* (*Yf*) [21]. "White cocoon" is presumed to be expressed by additive cocoon color discoloration by a combination of loss-of-function haplotypes of these loci. Although loss of *GH1G5* is not necessarily essential for cocoon whiteness, given the high conservation rate of loss-of-function haplotypes and the fact that knockout of the gene reduces the cocoon total flavonoid content to less than half (Fig 2F and 5F), the gene still makes an important contribution to cocoon discoloration in the domestic silkworm. Interestingly, the knockout of *GH1G5* increased the cocoon weight (S7 Fig). Because endoreduplication is an important step in silkworm silk gland development [48], interference with the replication step by quercetin interaction with DNA may explain the relationship between *GH1G5* and silk yield [49]. Although it will be difficult to confirm this hypothesis because cocoon weight is a complex quantitative trait dependent on silkworm physiology and behavior, elucidation of the relationship between quercetin and silk yield may provide clues to further improve silkworm protein productivity.

## Materials and methods

### Insect materials

All domestic silkworm strains used in the present study were maintained at the National Agriculture and Food Research Organization (NARO, Japan). p50T, the strain used for the functional analysis of *GH1G5*, is almost genetically identical to p50, because it is a descendant of p50 which had undergone repeated passage using a single pair to improve homozygosity for referential use. The silkworms were reared on a commercial artificial diet (SilkMate PM; Nosan, Kanagawa, Japan) or fresh mulberry leaves under a controlled environment (12-h light/dark photoperiod, 25°C). Insects reared on fresh mulberry leaves were used for measurement of the flavonoid contents of the cocoons of p50 and J01 (Fig 1A and 1B), the scoring for QTL analysis (Fig 1C), and measurement of the flavonoid contents of the cocoons of p50T and *GH1G5*-knockout mutants (S5 Fig). All individuals were female except those from which genomic DNA and RNA-seq data were derived.

### Flavonoid content measurement

After shredding the cocoons into 2–3-mm squares, flavonoids were extracted from the pieces with MeOH–$H_2O$ (7:3, V/V) at 60°C for 2 h. The extraction was repeated twice for thorough extraction. The flavonoid content in the solution was derived from the absorption at 365 nm, which is highly correlated with flavonoid content ($r$ = 0.95) [19]. To use flavonoid content as an input for QTL analysis, the raw absorbance values were used as relative phenotype scores. The tissues and organs for measuring flavonoid content were sampled from sixth-day final instar larvae reared on a commercial artificial diet, rinsed well with PBS buffer (pH 7.4) (Takara Bio, Shiga, Japan), and stored at −80°C. To inhibit melanization in hemolymph, immediately after collection 0.5 M sodium dimethyldithiocarbamate (Fujifilm, Tokyo, Japan) was added to reach 0.5% by volume. Quantification of flavonoid content in the samples was performed according to a previous report [15] with some modifications. In brief, 30 μL of sample was injected into an LC-10A HPLC system equipped with an SPD-M10AVP photodiode array detector (both Shimadzu Co., Kyoto, Japan) and separated with a SunFire C18 column (150 × 3.0 mm i.d.; Waters, Massachusetts, USA) at a flow rate of 1.0 mL/min. Spectrum

analysis and flavonoid peak assignment were performed with the CLASS-VP HPLC software (Shimadzu Co.). Flavonoids were quantified using quercetin 5, 4´-di-*O*-glucoside as the reference compound.

## QTL analysis

The intercross $F_2$ population of p50T and J01 used in the present study were the same samples as those sequenced by double-digest restriction-associated DNA sequencing and used to generate linkage maps in our previous study [34]. The sequencing data of 102 female $F_2$ individuals are available in the Sequence Read Archive under the BioProject accession number PRJDB13956. The raw values of absorbance indicating the relative cocoon flavonoid content for each individual are summarized in S1 Table. The method for genotyping, marker selection, and construction of linkage maps is described in detail in our previous report [34]. Briefly, OneMap v2.8.2 [50] was used to construct linkage maps with an LOD score of 3, with the genotype of the $F_2$ population output from the script "ref_map.pl" in STACKS v1.48 [51] as the input. The p50T genome assembly was used as the reference for read mapping [23]. The linkage map consisted of 1038 markers covering a total genetic length of 945.36 cM, with an average marker density of 0.936 cM. QTL analysis was performed using the R/qtl v1.46.2 package [52], with the relative flavonoid content scores in the cocoons and the genetic position of markers linked to the physical position on the p50T genomic sequence as inputs. Using the "calc.genoprob" function in R/qtl, the probabilities of the true underlying genotypes were calculated with a step size of 1 cM and an assumed genotyping error rate of 0.05. QTL detection was performed by composite interval mapping using the function "cim" with the following parameters: method = "hk", n.marcovar = 3, window = 10. The LOD significance threshold for detecting QTL was calculated by a permutation test of 1000 trials. Approximate Bayesian 95% credible intervals were calculated using the function "bayesint". The positions of the nearest markers outside of the boundaries of a confidence interval were defined as the ends of the interval. PVEs and additive effects of the three significant QTLs were estimated using the function "fitqtl" with the parameter: formula = y ~ Q1 + Q2 + Q3 + Q1 * Q2 * Q3.

## Phylogenetic analysis

GH1 proteins in *B. mori* were identified using InterPro annotation (InterPro entry: IPR001360, Glycoside hydrolase family 1) [53]. An InterPro annotation result for the p50T predicted protein sequences is available in KAIKObase [54] (https://kaikobase.dna.affrc.go.jp/index.html). The related proteins in Holometabola insects were collected by BLASTp search using the 21 *B. mori* GH1 proteins as the query. Genes with e-values less than 1E-50 were used for the analysis. InsectBase 2.0 (http://v2.insect-genome.com/) [55] was used for picking related proteins in *Drosophila melanogaster*, *Tribolium castaneum*, and *Apis mellifera*. *Arabidopsis* thioglucoside glucohydrolase 1 (TGG1) was used as an outgroup. The query and these sequences were aligned using Clustal Omega v1.2.4 [56]. Tree construction was performed by the neighbor-joining method using MEGA 11 [57] with the number of bootstrap trials set to 100. The phylogenetic tree of lepidopteran orthologs of the *Gd* glycoside hydrolases was constructed by the maximum likelihood method using RAxML-NG v1.1 [58] with the following parameters:—bs-trees 100,—model LG+I+G4 (Fig 4A). The best-fit phylogenetic analysis model was suggested using ModelTest-NG v0.1.7 [59]. Gene models of 12 lepidopteran insects available in InsectBase 2.0 were used (*Papilio xuthus*, *Megathymus ursus*, *Lycaena phlaeas*, *Limenitis camilla*, *Chilo suppressalis*, *Biston betularia*, *Pheosia gnoma*, *Dendrolimus punctatus*, *Bombyx mori*, *Bombyx mandarina*, *Antheraea yamamai*, *Deilephila porcellus*). Each of the models was predicted based on high-quality genome sequences using long-read sequencing,

and their BUSCO [60] scores were all above 94%. The tree included a total of 90 lepidopteran protein sequences, which were inferred to belong to the same ortholog groups as the nine group-G glycoside hydrolases identified by OrthoFinder [32] v2.5.2; Rat LPH (NCBI Refseq accession: NP_446293) as an outgroup.

## Gene editing

CHOPCHOP v3 was used to identify target sites [61]. Each 20-nucleotide guide sequence was unique on the p50T genomic sequence [23]. The construction of DNA templates for the synthesis of single-guide RNA (sgRNA) was performed according to a previous report [62]. Briefly, the DNA template, which included the T7 promoter, sgRNA target with PAM motif, and tracrRNA-complemental sequence, was synthesized by template-free PCR. A MEGAshortscript T7 Transcription Kit (Thermo Fisher Scientific, Waltham, Massachusetts, U.S.A.) was used for *in vitro* sgRNA transcription and template DNA digestion. A Guide-it IVT RNA Clean-Up Kit (Takara Bio) was used to clean up the sgRNA solution. Alt-RS.p. HiFi Cas9 Nuclease V3 solution (Integrated DNA Technologies, Coralville, Iowa, U.S.A.) and the sgRNA were mixed with distilled water to final concentrations of 500 ng/μL and 50 ng/μL. The Cas9 nuclease and sgRNA mixture was incubated for 1 h at room temperature to allow the ribonucleoprotein to form. The ribonucleoprotein solution was micro-injected into non-diapause eggs of the p50T strain within 8 h after oviposition, in accordance with a previous report [63]. Non-diapause treatment was performed by incubating eggs under 17°C short-day conditions (12-h light/dark photoperiod) until hatching. Adult moths of G0 individuals (injected generation) were crossed with wild-type moths. Individuals harboring identical heterozygous mutant haplotypes were identified by high-resolution melting (HRM) analysis using genomic DNA extracted from G1 generation molt shells harvested from cocoons. PCR for HRM analysis was performed using a KAPA HRM Fast PCR Kit (Roche, Basel, Switzerland) on a LightCycler 96 System (Roche) according to the manufacturer's instruction. Genomic DNA extraction from molt shells was performed by homogenizing the shells in a typical SDS-based lysis buffer followed by isopropanol precipitation. G2 individuals with a homozygous mutant haplotype were identified and crossed by the same method to establish the knockout lines. To confirm the presence of a mutation PCR products which included the target site were sequenced using a BigDye Terminator v3.1 Cycle Sequencing Kit and an Applied Biosystems 3130xl Genetic Analyzer (Thermo Fisher Scientific). The primer sequences used are listed in S7 Table.

## Photographing of dissected samples

Sixth-day fifth instar larvae reared on the artificial diet were used. Photographs were collected with a DP74 camera (Olympus, Tokyo, Japan) attached to an SZX16 microscope (Olympus). To inhibit melanization of collected hemolymph, 0.5 M sodium dimethyldithiocarbamate (Fujifilm) was added immediately after collection to reach 0.5% by volume. To remove fluorescent hemolymph before imaging, silk glands and whole-body samples from which silk glands had been removed were rinsed well with PBS buffer (pH 7.4) (Takara Bio). A fluorescence light source U-HGLGPS (Olympus) and ultraviolet filter set (excitation: 330–385 nm, emission: 420 nm longpass) (Olympus) were used for imaging under ultraviolet irradiation. Because of the loss of resolution and optical noise when the field of view was made large enough to fit whole bodies, these samples were photographed in segments and then the pictures were combined. In the photographs shown in Fig 2, no post-editing of any kind was performed other than cropping.

### Determination of *GH1G5* open reading frame sequence

Total RNA was extracted from the midgut of sixth-day final instar larvae using ISOGEN (Nippon Gene, Tokyo, Japan) according to the manufacturer's instructions. Contaminating DNA in the RNA solution was digested using RNase-Free DNase Set (Qiagen, Hilden, Germany). The isolated RNA was purified using an RNeasy Kit (Qiagen). Equal volumes of RNA solutions from five individuals adjusted to the same concentration were mixed to obtain a single bulk sample. Reverse transcription was performed using ReverTra Ace qPCR RT Master Mix (Toyobo, Osaka, Japan) according to the manufacturer's instruction, and the final concentration of the RNA was adjusted to 10 ng/μL. KOD One PCR Master Mix -Blue- (Toyobo) was used for PCR according to the manufacturer's instructions. The conditions of preincubation, denaturation, annealing, and extension were set to 30 s at 95˚C, 10 s at 98˚C, 10 s at 55˚C, and 5 s at 68˚C, for 42 cycles. Amplification products were purified using a QIAquick PCR Purification Kit (Qiagen) and cloned into a vector using a Zero Blunt TOPO PCR Cloning Kit (Thermo Fisher Scientific). Sequencing was performed as described in the subsection above, Gene editing. The primer sequences are listed in S7 Table.

### Isoform identification and expression analysis

RNA-seq data from the midgut and other organs of third-day fifth instar larvae of p50T and J01 were previously published by our research group [30,34]. The sequencing reads were trimmed by fastp v0.20.0 [64] with the following parameters: -q 20 -n 5 -l 100. Read mapping to the p50T genome assembly and transcript isoform identification were conducted using HISAT2 v2.1.0 [65] and StringTie2 v2.2.0 [66] with default parameters. Salmon v1.5.2 [67] was used to calculate transcripts per kilobase million scores. The predicted transcript sequence of p50T was used as the reference [23].

### Assay for hydrolytic activity of the midgut

Each midgut of four independent fifth-day final instar larvae reared on the commercial artificial diet was collected, rinsed well with PBS buffer (pH 7.4) (Takara Bio), and immediately stored at −80˚C. Three times the weight of 10 mM phosphate/1 mM PMSF buffer (pH 7.5) as the sample was added prior to homogenization, which was thoroughly performed on ice, using a metal homogenizer Physcotron with a microshaft NS-4 (Microtec, Chiba, Japan). Three times the weight of a 20 mM phosphoric acid (pH 5.5) solution as the homogenate was add prior to hydrolysis, which was performed independently with each of the three substrates, rutin, Q3MG and Q3G, at a concentration of 2 mM for 4 h at 37˚C. The enzyme-independent degradation of the three quercetin glycosides and quercetin in the reaction was first examined by deactivating the homogenate by adding three volumes (V/V) of methanol to the reaction liquid. We measured the change in amounts of quercetin from the substrates at 0 h and 4 h after the reaction which confirmed they were stable in the condition. Control incubations in which the substrates were omitted were also run for calibration taking into account flavonoid content in the tissue. The reaction was stopped by adding three volumes (V/V) of methanol. After centrifugation at 20,000$g$ for 10 min, the quercetin content of the supernatant was quantified with the HPLC system as described in the section "Flavonoid content measurement." Quercetin, rutin, and Q3G were obtained from Extrasynthese (Lyon, France). Q3MG was obtained from Merck (Darmstadt, Germany). Hydrolytic activity for the quercetin glycosides is expressed as nanomoles of quercetin produced per min per mg protein. The protein concentration in the enzyme preparations was measured with a commercial assay kit (Coomassie Plus, Thermo Fisher Scientific).

## Dietary administration of quercetin or rutin

Silkworm larvae were reared on the commercial diet from hatching to the third ecdysis, then the fourth instar larvae were fed with a diet containing 25% mulberry leaf powder [68]. The newly molted fifth instar larvae were reared on semi-synthetic diets supplemented with quercetin or rutin, which did not contain mulberry leaf powder (S3 Table). Diets were supplemented with flavonoids in approximately equimolar (0.1 mmol/100-g dry diet) amounts, and chlorogenic acid was added as a feeding stimulant. Soybean meal (*Soya flour FT*; Nisshin Oil-liO Group,. Ltd., Tokyo, Japan) was washed twice with five volumes of 90% (V/V) ethanol to remove deterrent substances and allowed to dry naturally before use in the diets.

## Genomic comparison of the *Gd* locus

Dot-plot analysis was performed using FlexiDot v1.06 [69] with the following parameters: -p 1 -f 1 -A 2 -E 100. The predicted protein sequence of J01 *GH1G5* (*BMN13127*) presented here (S8B Fig) was modified from the original model; notably, the incorrect prediction of exon 1 was modified according to *KWMTBOMO12227*, and the J01 transcript sequence was obtained by using the original RNA-seq data [34]. DNA extraction from the whole body of a male larva of the domestic silkworm strain Kosetsu was performed by a standard phenol-chloroform-based method. Whole genomic sequencing was performed by an outsourcing service (Macrogen Japan Corp., Kyoto, Japan) using an Illumina NovaSeq 6000 Sequencing System (Illumina Inc., San Diego, USA). The sequencing generated 50.4 Gb of paired-end reads with a length of 151 bp. Genome assembly of the Kosetsu strain was performed using MaSuRCA v4.0.9 [70] with pseudomolecule sequences of chromosomes 15, 20, and 27 of p50T [23] which contain the QTLs as the assembly reference. The parameter defining 20 times genomic size "JF_SIZE" was set to 9,000,000,000. The input genomic sequencing data of the Kosetsu strain was trimmed by fastp v0.20.0 [64] before assembly with the following parameters: -q 20 -n 5 -l 100.

## PCR-based genotyping

The genomic DNA of each strain was extracted from 10 pairs of silk glands of final instar larvae by a standard phenol-chloroform-based method. The final concentration of genomic DNA was adjusted to 0.5 ng/μL. KOD One PCR Master Mix -Blue- (Toyobo) was used for PCR according to the manufacturer's instructions. The conditions of preincubation, denaturation, annealing, and extension were set to 30 s at 95˚C, 10 s at 98˚C, 10 s at 60˚C, and 45 s at 68˚C, for 32 cycles. The primer sequences are listed in S7 Table.

## Reverse-transcription PCR

Reverse transcription was performed as described in the subsection above, determination of *GH1G5* open reading frame sequence. RT-PCR (Fig 5E) was performed using KAPA HiFi HotStart ReadyMix (Roche) under conditions that resulted in 1% of the concentration of the original cDNA solution. The conditions of preincubation, denaturation, annealing, and extension were set to 180 s at 95˚C, 5 s at 98˚C, 10 s at 60˚C, and 60 s at 72˚C, for 28 cycles. The primer sequences are listed in S7 Table.

## Detecting gene duplication events

Detecting gene duplication events in the lepidopteran glycoside hydrolases was performed using OrthoFinder v2.5.2 [32] together with the ortholog-group inference for the phylogenetic analysis. Proteins annotated as "Glycoside hydrolase superfamily (IPR017853)" or with the

functional annotation "hydrolase activity, acting on glycosyl bonds (GO:0016798)," were considered to be glycoside hydrolases.

## Other bioinformatics tools

Handling sequences: Seqkit v2.2.0 [71]; statistical calculations: R v4.0.3 (R Foundation for Statistical Computing, Vienna, Austria); illustrating graphs: ggplot2 v3.4.1 [72] and ggpubr v0.6.0 (https://CRAN.R-project.org/package=ggpubr); alignment visualization: SnapGene Viewer (https://www.snapgene.com/); prediction of signal peptide domains: SignalP v6.0 [73]; illustrating structural formulae: Ketcher v2.8.0 (https://github.com/epam/ketcher); genomic browsing: Integrative Genomics Viewer v2.12.3 [74]; phylogenetic tree editing: Interactive Tree Of Life v5 [75].

## Supporting information

**S1 Fig. Photographs of representative cocoons of the wild silkworm, *Bombyx mandarina*.** A fresh mulberry leaf and cocoons of the wild silkworm in a bright field (left) and irradiated with ultraviolet A in a dark field (right). The cocoons under ultraviolet irradiation exhibit fluorescence characteristic of quercetin-5-*O*-glucoside and quercetin-5,4´-di-*O*-glucoside, the major quercetin metabolites in the silkworm tissues and cocoon [14,15,29]. The cocoons were collected in June 2023 at Tsukuba, Ibaraki, Japan. Bar = 10 mm.
(TIF)

**S2 Fig. Phylogenetic tree of GH1 proteins in *B. mori* and other Holometabola insects.** Values on the nodes represent the bootstrap score (trials = 100). Unreliable nodes with bootstrap values under 50 are shown as multi-branching nodes. *Arabidopsis* thioglucoside glucohydrolase 1 (TGG1) was used as the outgroup.
(TIF)

**S3 Fig. Similarity of the group G glycoside hydrolase proteins in *B. mori*.** (A) Alignment of the group G glycoside hydrolase proteins in *B. mori* using Clustal Omega. Predicted signal peptides are highlighted by red lines. The threshold for identity shading was set to 80%. (B) Sequence identity and similarity of the group G glycoside hydrolase proteins in *B. mori*. Similarity of amino acid residue property is determined according to the groups of strongly similar properties described at Clustal Omega FAQ (https://www.ebi.ac.uk/seqdb/confluence/display/THD/Help+-+Clustal+Omega+FAQ).
(TIF)

**S4 Fig. Expression profile of the candidate genes within the *Gd* locus in third-day final instar larva of the p50T strain.** All organs and tissues, except for ovaries, were taken from male larvae. Data are means of n independent biological replicates ± SD. SG, silk gland; TPM, transcripts per million.
(TIF)

**S5 Fig. Total flavonoid content in cocoons of the p50T strain and *KWMTBOMO12227*-knockout lineages reared on fresh mulberry leaves.** Data are means of n independent biological replicates ± SD.
(TIF)

**S6 Fig. RNA-seq read alignment to the genomic sequence of *GH1G5*.** (A) Regions around the translation start site (up) and stop site (down) of the cloned transcript sequence of *GH1G5*. The annealing sites of primers used for cloning are highlighted by black lines. (B) Mapping results of the midgut-derived RNA-seq reads and *GH1G5* transcript isoforms visualized with

Integrative Genomics Viewer. Coverage on each exon and splice junction of three independent biological replicates is illustrated on tracks R1–3. (C) Coverage mean of each base of exon 10 and the 15-bp extension region. Data are means of n independent biological replicates ± SD.
(TIF)

**S7 Fig. Weight of pupae and cocoons of p50T and *GH1G5* knockout mutants.** (A) Weight of the pupae. (B) Weight of the cocoons. The values above the graph are *p*-values calculated by Student's *t*-test. Crosses inside the boxes indicate means of n independent biological replicates.
(TIF)

**S8 Fig. Comparison of *GH1G5* sequences between the p50T and J01 strains.** (A) Dot-plot analysis of the genomic region of exons 5, 6 and the intronic region between them (intron 5) of *GH1G5* from the p50T and J01 strains. Green dots indicate inverted regions. (B) Alignment of predicted protein sequences of GH1G5 from the p50T and J01 strains using Clustal Omega [56]. (C) Deletion in the J01 *GH1G5* transcript confirmed using Sanger sequencing.
(TIF)

**S9 Fig. Alignment of GH1G5-like proteins in Macroheterocera species.** (A) Whole alignment of 101 GH1G5-like proteins in Macroheterocera species. Amino acid residues identical to GH1G5 at positions with >99% conservation are highlighted by blue. The 100 homologous proteins were collected by a BLASTp search using the amino acid sequence of GH1G5 from sequence data of each protein from *Bombyx mori* (KWMT), *Antheraea yamamai* (Ayam), *Deilephila porcellus* (Dpor), *Dendrolimus punctatus* (Dpun) and *Pheosia gnoma* (Pgno) with a cut-off criterion e-value of <1E-100. The sequence data of *B. mori* were obtained from KAIKObase (https://kaikobase.dna.affrc.go.jp/index.html) and those of *A. yamamai*, *D. porcellus* and *P. gnoma* were obtained from InsectBase 2.0 (http://v2.insect-genome.com/). The sequences were aligned using Clustal Omega. (B) Magnified view of the J01-type deletion region of the alignment. Descriptions of the consensus symbols indicating the conservation level of amino acid residues are available at the Clustal Omega FAQ (https://www.ebi.ac.uk/seqdb/confluence/display/THD/Help+-+Clustal+Omega+FAQ). In brief, ':' indicates residues harboring a strongly similar physicochemical property to GH1G5, and '.' indicates residues harboring a weakly similar physicochemical property to GH1G5.
(TIF)

**S10 Fig. Genotyping of the *Gd* locus of Japanese or Chinese local white-cocoon strains.** Haplotypes of the strains were determined by PCR genotyping. P/P, only the p50T genotype was detected; P/J/K, all p50T, J01, and Kosetsu genotypes were detected; J/J, only the J01 genotype was detected; J/K, both the J01 and Kosetsu genotypes were detected; K/K, only the Kosetsu genotype was detected; U, unknown, nothing detected. The primer set amplifying the genomic region from exons 1 to 2 of *KWMTBOMO14639* (*rp49*) was used as a control (predicted fragment length = 1548 bp). The straight grey dotted lines connect band markers of known size (M) which were applied at both ends of the samples. These lines represent sizes of 4000, 200, 500, and 1500 bp, from top to bottom. "JL" and "CL" stand for "Japanese local" and "Chinese local". The colors indicate the color of the cocoons.
(TIF)

**S1 Table. Information of the F$_2$ individuals, parents and phenotypic score used for the QTL analysis.**
(XLSX)

**S2 Table. QTLs associated with flavonoid content in cocoons.**
(XLSX)

**S3 Table. Composition of the semi-synthetic diets supplemented with quercetin or rutin.**
(XLSX)

**S4 Table. Orthogroups of glycoside hydrolases in the 12 species of Lepidoptera and numbers of detected gene duplication events.**
(XLSX)

**S5 Table. Domestic silkworm glycoside hydrolases.**
(XLSX)

**S6 Table. Summary of the Japanese and Chinese local silkworm strains examined.**
(XLSX)

**S7 Table. Primers used in the experiments.**
(XLSX)

**S1 Data. Data underlying the reported results.**
(XLSX)

## Acknowledgments

We thank Ms. Mayuko Sakato for her technical assistance; Sae Furuhashi, Toshihiko Misawa, Kaoru Nakamura, Keita Hiruma, Koji Hashimoto, and Eiji Okada for their support in the silkworm rearing; Hiroki Sakai for giving instruction on the CRISPR-Cas9 system construction; Nobuto Yamada for giving instruction on the microinjection. The computation was partly performed using the supercomputer of the Agriculture, Forestry and Fisheries Research Information Technology Center of the Japanese *Ministry of Agriculture, Forestry and Fisheries*.

## Author Contributions

**Conceptualization:** Ryusei Waizumi, Chikara Hirayama, Tetsuya Iizuka.

**Data curation:** Ryusei Waizumi.

**Formal analysis:** Ryusei Waizumi.

**Funding acquisition:** Tetsuya Iizuka, Hideki Sezutsu.

**Investigation:** Ryusei Waizumi, Chikara Hirayama, Tetsuya Iizuka, Seigo Kuwazaki, Akiya Jouraku.

**Project administration:** Tetsuya Iizuka.

**Resources:** Shuichiro Tomita, Tetsuya Iizuka, Takuya Tsubota, Kakeru Yokoi.

**Supervision:** Shuichiro Tomita, Kimiko Yamamoto, Hideki Sezutsu.

**Visualization:** Ryusei Waizumi.

**Writing – original draft:** Ryusei Waizumi.

**Writing – review & editing:** Ryusei Waizumi, Chikara Hirayama, Shuichiro Tomita, Akiya Jouraku, Takuya Tsubota.

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
