## [Decision Letter · Decision Letter 0]

8 Sep 2023

Dear Dr Waizumi,

Thank you very much for submitting your Research Article entitled 'A major endogenous glycosidase mediating quercetin uptake in *Bombyx mori*' to PLOS Genetics.

The manuscript was fully evaluated at the editorial level and by independent peer reviewers. The reviewers appreciated the attention to an important problem, but raised some substantial concerns about the current manuscript. Based on the reviews, we will not be able to accept this version of the manuscript, but we would be willing to review a much-revised version. We cannot, of course, promise publication at that time.

If you decide to revise the manuscript for further consideration at PLOS Genetics, please aim to resubmit within the next 60 days, unless it will take extra time to address the concerns of the reviewers, in which case we would appreciate an expected resubmission date by email to plosgenetics@plos.org.

We are sorry that we cannot be more positive about your manuscript at this stage. Please do not hesitate to contact us if you have any concerns or questions.

Yours sincerely,

Takaaki Daimon

Academic Editor

PLOS Genetics

Gregory P. Copenhaver

Editor-in-Chief

PLOS Genetics

Reviewer's Responses to Questions

**Comments to the Authors:**

Reviewer #1: This manuscript presents a comprehensive overview of the discovery of a third locus within the silkworm that plays a crucial role in the uptake of quercetin from mulberry leaves. This specific locus consists of a cluster of genes arranged in tandem, all of which encode glycoside hydrolase family 1 proteins. Knockout of one of these genes resulted in a noticeable decline in the uptake of quercetin, particularly of rutin. Notably, the researchers demonstrated that mutations occurring in this gene have been selected for in various silkworm strains due to their association with enhanced silk production. In my assessment, I found this paper to be highly captivating and the experiments to be very well designed. Presented below are a few comments aimed at further enhancing the quality of the paper.

Reading through the paper, I was not really sure that the glycosidases encoded by the gene cluster on chromosome 20 were actually “glycoside hydrolase family 1 (GH1)” proteins (in line with the CAZy nomenclature). I would recommend that the authors address this point more explicitly in the revised version of their manuscript. For example on page 7, line 132, the term “beta-glycosidase family 1” is misleading and does not follow the CAZy nomenclature naming carbohydrate-active enzymes.

On page 15, lines 217-219, I am of the opinion that the entity you identify as "a hydrophobic transmembrane domain in the first N-terminal 20 amino acid residues" could potentially represent a signal peptide. Considering the context that LQGH1 is linked to the apical membrane of midgut cells, oriented towards the exterior of the cell, the requirement for a signal peptide becomes apparent to facilitate such subcellular localization. In light of this, I recommend that the authors explore this possibility using the prediction program SignalP v6.0 (https://services.healthtech.dtu.dk/services/SignalP-6.0/), which has been optimized to more effectively discriminate between signal peptides and hydrophobic transmembrane domains.

I am convinced that LQGH1 plays a pivotal role in quercetin uptake through deglycosylation of quercetin glycosides. This conviction stems from the outcomes of the presented knockout experiments coupled with the hydrolysis assays. However, I found it somewhat disappointing that the authors did not attempt to express LQGH1 and the other GH1 proteins from the cluster in a heterologous system. On the other hand, considering the substantial volume of data presented in this study, I think that this absence should not necessarily hinder the acceptance of the paper for publication. I am hopeful that the inclusion of such data could be a valuable addition to a subsequent paper on this subject.

In my view, a notable strength of this paper lies in the authors' commitment to investigating the potential existence of naturally occurring mutations at the LQGH1 locus within silkworm populations extensively employed for silk production. The authors successfully identified multiple distinct mutations at the LQGH1 locus in silkworm populations from both Japan and China. Importantly, these mutations were correlated with an increased silk production. Altogether, I strongly believe that a polished version of this paper, taking into account my comments, would fit very nicely in PloS Genetics.

Minor comments:

Page 4, line 56: Replace “high” with “higher”.

Page 5, line 82: replace ”enzymic activity” with “enzymatic activity”.

Page 6, line 106: replace “gradated range” with “graduated range”.

Reviewer #2: Uploaded as an attachment

Reviewer #3: The manuscript describes the discovery of a b-glucosidase that is involved in the sequestration of plant flavonoids in cocoons of the silkworm Bombyx mori. This study combined forward and reverse genetic approaches with chemical and biochemical analyses resulting in an impressive dataset. The results support the hypothesis that deglycosylation of flavonoid glycosides (especially rutin) by LQGH1, is essential step for their uptake probably via passive absorption of the hydrolysis product quercetin. While I really enjoyed reading the study, I have major concerns about the in silico amino acid sequence analysis, enzyme assays, the lack of LQGH1 sequence verification from both strains using Sanger sequencing, and conclusions on the localization and the role of unequal crossing over in generating the different haplotypes described in this study. Once these concerns are addressed, this study will be an excellent contribution to the field of molecular insect-plant interactions and the versatile roles of b-glucosidases in insects. The authors strongly emphasize their findings in the context of silkworm domestication and breeding, which I think could be a bit more balanced in the discussion with regard to implications for insect-plant coevolutionary interactions.

My detailed comments and suggestions are listed below:

Abstract l 19: “here we identified and characterized a glycosidase”: this statement implies that the enzyme was heterologously expressed and biochemically characterized. As this was not done, please rephrase. In general, to account for the fact that the enzyme was not heterologously expressed and characterized, please tone down the conclusions on the function.

Abstract l 24: “unequal crossing over hotspot”: I have difficulties to follow the conclusion that all the different variants result from unequal crossing over. Are there no other possible explanations?

Introduction line 84: quercetin glucosidases or quercetin glycosides?

Results line 132: LPH belong to the glycoside hydrolase (GH) family 1

Line 146: correct to: Malpighian tubules

Lines 173-175: was it possible to obtain heterozygous knock-out mutants? If yes, what was their phenotype?

Results lines 205 ff: I have several major concerns regarding the sequence analysis and enzyme assays:

1) Amino acid sequence similarity between mammalian LPH and LQGH1: both proteins are predicted to belong to the GH1 family based on their amino acid sequence, so it is expected that their sequences are similar. In my opinion, this statement is unnecessary but if the authors have important reasons to keep the statement, they should at least give the percentages of sequence identity/similarity.

2) Similarly, an orthologous relationship of LPH and LQGH1 is not expected and is impossible to deduce with phylogenetic analyses as is demonstrated by the lack of statistical support for most nodes, and the fact that only gene clusters from the same species are supported at all. Similar results have been obtained in other studies on insect GH1s, which should be included in the disucssion. I think a more thorough analysis of the GH1 gene family in Bombyx (e.g., the number of predicted genes, their phylogenetic relationships and sequence similarities, in particular within the cluster that harbors LQGH1) that is then placed in the larger context of Lepidopteran GH1s as shown in Fig. S11 would be more informative and meaningful.

3) The authors state that LQGH1 contains a hydrophobic transmembrane domain at the N-terminus. I am unfamiliar with the tool that the authors used to predict this and I also have difficulties to interpret the graph shown in Fig. S4. Many insect GH1 are secreted proteins that contain an N-terminal secretion signal. Considering that at least some GH1s are involved in digestion, it is also expected that they are secreted into the gut lumen. I repeated the sequence analysis using several programs that are widely used to predict the presence of a signal peptide and transmembrane domains in proteins as well as protein localization (SignalP, TMHMM, DeepLoc). All these programs are available online and are free of charge. The results clearly show that the enzyme has an N-terminal secretion signal, no transmembrane domain, and is located extracellularly. It is thus very unlikely that LQGH1 is membrane-anchored and instead is secreted into the gut lumen.

4) Based on this result I have additional concerns regarding the enzyme assays with gut homogenates. In general, the description of the methods is not sufficient. Please specify how the dissected guts were homogenized, if samples consisted of pooled guts and if the replicates are biological or technical replicates. Are homogenate and insoluble fraction samples derived from the same initial homogenate? Was the insoluble fraction dissolved again in the same volume?

5) In case of the enzyme assays the authors are missing an important background control, the substrate itself. Instead of using protein without substrate, boiled protein with substrate should be used, or at least controls without protein but including the substrate. This is an important control because very often you will find some contamination of the substrate. Were the assays carried out with a substrate mixture or individual substrates? Why was the insoluble fraction not tested for the mutant GD2?

6) As explained above, the comparison of homogenate with the insoluble fraction is not appropriate. Instead, the homogenate should be separated into soluble and insoluble fraction for enzyme assays.

Fig. 3: The structures of rutin and Q3G are wrong and have to be corrected. Please carefully check all structures again.

Line 300: two sites within exon 5?

Line 303: It is not really clear how the insertion of intron sequences into an exon leads to a protein sequence that lacks 31 amino acids. Can this be shown graphically?

The difference in amino acid sequence should also be confirmed by amplifying and Sanger-sequencing the full-length sequence from cDNAs from both strains. If I understand correctly, the sequence was only confirmed in strain p50T based on RNA-Seq data but not in the J01 strain.

Line 328: what exactly is meant with “weaker signal”?

Line 332: What is the exact evidence for the genomic changes being the result of unequal crossing over?

Fig. 4A: the dot plot analysis is not easy to understand. What is the significance of the different colors (green versus black)?

Line 370: uptake from the midgut?

Line 379-380: it would be indeed great if functional studies with recombinant LQGH1 and other enzymes encoded in the gene cluster could be included in this study

Line 395-396: I would be more careful with the conclusion that LQGH1 has conserved rutin hydrolytic activity, I don’t think that the available data are sufficient to support this conclusion

Line 401: why is it a peculiar function? Consider to rephrase.

Line 417-418: please rephrase, “crisis of loss”. Again, I am not convinced that all the variants are explained by unequal crossing over.

Line 418-419: What is meant by “remained conserved”? The enzymatic activity of putative orthologs has not been tested. Please tone down the conclusions.

Line 425: I think it would helpful to have a cartoon showing the proposed steps in quercetin glycoside metabolism, transport, and excretion in Bombyx

**Have all data underlying the figures and results presented in the manuscript been provided?**

Reviewer #1: Yes

Reviewer #2: Yes

Reviewer #3: **No: **I did not find a spreadsheet containing numerical data underlying graphs and statistics.

PLOS authors have the option to publish the peer review history of their article (what does this mean?). If published, this will include your full peer review and any attached files.

Reviewer #1: No

Reviewer #2: **Yes: **Marian R. Goldsmith

Reviewer #3: No

---

## [Decision Letter · Decision Letter 1]

27 Nov 2023

Dear Dr Waizumi,

Thank you very much for submitting your Research Article entitled 'A major endogenous glycoside hydrolase mediating quercetin uptake in *Bombyx mori*' to PLOS Genetics.

The manuscript was fully evaluated at the editorial level and by independent peer reviewers. The reviewers appreciated the attention to an important topic but identified some concerns that we ask you address in a revised manuscript.

We therefore ask you to modify the manuscript according to the review recommendations. Your revisions should address the specific points made by each reviewer.

Yours sincerely,

Takaaki Daimon

Academic Editor

PLOS Genetics

Gregory P. Copenhaver

Editor-in-Chief

PLOS Genetics

Comments from the Associate Editor:

The author has nicely revised the mauscript.

In this final round of review, I have some additional comments that would strengthen the manuscript.

1. Fig. 6

I think the authors could improve Fig. 6 to make it clearer and more comprehensive.

For example, the process of deglycosylation and subsequent re-glycosilation of plant-derived quercetin/flavonoids should be shown more clearly.

Also, the process of uptake and excretion of flavonoids, i.e., from the gut lumen to the silk gland lumen, in the Bombyx could be described in the figure.

2. other loci

There are other loci involved in the flavonoid cocoon trait.

The author may want to include such loci, e.g., Green n, Green a, and Lg/P5CR in Fig. 6.

Please also note that reviewer #2 has attached the pdf file with the possible changes to the manuscript.

Finally, if the authors agree, it would be nice to acknowledge the three anonymous reviewers who provided constructive and valuable suggestions in the Acknowledgments section.

Reviewer's Responses to Questions

**Comments to the Authors:**

Reviewer #1: I am satisfied by the changes made by the authors. The manuscript has been substantially improved. I see no major reason that should prevent acceptance for publication. I have a few minor comments, mostly typos, that need to be attended to.

Minor comments:

- page 7, line 130. Title of the legend of Fig. 1, replace "Quantitive" by "Quantitative".

- page 10, line 194. Add "in" before "midgut".

- page 15, line 262. Replace "glycosidses" by "glycosidases".

- page 24, line 423. Replace "retainend" by "retained".

- page 33, line 598. Write "Arabidopsis" in italic.

- page 37, line 678. Replace "expressional" by "expression".

- page 37, line 691. Replace "throughly" by "thoroughly"

- page 37, line 692. Replace "homozenizer" by "homogenizer".

- page 37, line 696. Do you mean "Previously" instead of "Precedently"?

- page 40, line 764. Maybe "Other bioinformatics tools" would sound better.

Reviewer #2: I have updated details of the review indicated below as an attachment (pdf edited with Adobe Acrobat)

The authors have addressed my comments and concerns well and thoroughly. This results in giving greater impact to the phylogenetic treatment by moving the essential data and reporting and discussion of it into the main text instead of assigning it to the supplementary text, and making numerous small changes to the language better to bring it in line with common English expression. Therefore I believe the revised treatment improves the potential impact of the research on a wider audience than initially and the readability of the text.

In my re-review I recommended a number of minor edits throughout the text to clarify the information and analysis I believe the authors intend to convey, and to improve the English expression a little more. Rather than extracting these edits into a line-by-line listing here I have marked them directly on the manuscript using Acrobat Reader which can be formatted to list embedded comments in a master list. If necessary I am willing to extract this information to standard line-by-line listings for editors’ and authors’ reference. However, I believe returning a marked up text along with my comments on the reasons for the proposed changes is a more efficient and useful way to convey this information.

Otherwise my general positive comments in the initial review stand so it seems unnecessary to repeat them here.

Reviewer #3: I am very happy with the revised manuscript and think the authors did an excellent job to address the comments. Below are some suggestions for minimal edits to the text. Line numbers refer to the version without track changes.

Line 173: I suggest to write “secreted enzymes” instead of “secretory enzymes”. The former term is used more frequently in the literature.

Line 221: please omit “absolutely”

Line 221 (and other places in the text): I suggest to write “predicted sequence” instead of “modeled sequence”

Line 683: correct “SrtingTie2” to “StringTie2”

**Have all data underlying the figures and results presented in the manuscript been provided?**

Reviewer #1: Yes

Reviewer #2: Yes

Reviewer #3: Yes

PLOS authors have the option to publish the peer review history of their article (what does this mean?). If published, this will include your full peer review and any attached files.

Reviewer #1: **Yes: **Yannick Pauchet

Reviewer #2: **Yes: **Marian R. Goldsmith

Reviewer #3: **Yes: **Franziska Beran

---

## [Editor Report · Decision Letter 2]

28 Dec 2023

Dear Dr Waizumi,

We are pleased to inform you that your manuscript entitled "A major endogenous glycoside hydrolase mediating quercetin uptake in *Bombyx mori*" has been editorially accepted for publication in PLOS Genetics. Congratulations!

Yours sincerely,

Takaaki Daimon

Academic Editor

PLOS Genetics

Gregory P. Copenhaver

Editor-in-Chief

PLOS Genetics

Comments from the reviewers (if applicable):

The authors nicely revised the manuscript.

It is now ready for publication.

**Data Deposition**

http://datadryad.org/submit?journalID=pgenetics&manu=PGENETICS-D-23-00886R2

**Press Queries**

---

## [Editor Report · Acceptance letter]

9 Jan 2024

PGENETICS-D-23-00886R2 

A major endogenous glycoside hydrolase mediating quercetin uptake in *Bombyx mori*

Dear Dr Waizumi, 

We are pleased to inform you that your manuscript entitled "A major endogenous glycoside hydrolase mediating quercetin uptake in *Bombyx mori*" has been formally accepted for publication in PLOS Genetics! Your manuscript is now with our production department and you will be notified of the publication date in due course.

With kind regards,

Anita Estes

PLOS Genetics

On behalf of:
